# Screening and Identification of Goat-Milk-Derived Lactic Acid Bacteria with Bacteriocin-like Activity and Probiotic Potentials

**DOI:** 10.3390/microorganisms11040849

**Published:** 2023-03-27

**Authors:** Zhenzhu Yang, Ling Ni, Wei Tian, Hai Chi

**Affiliations:** East China Sea Fisheries Research Institute, Chinese Academy of Fishery Sciences, Shanghai 200090, China

**Keywords:** lactic acid bacteria, bacteriocin-like substance, probiotics, antimicrobial characteristics, probiotic potential

## Abstract

In the current study, we screened 46 isolates of lactic acid bacteria (LAB) derived from goat milk for bacteriocin producers that can inhibit common foodborne pathogens (*Staphylococcus aureus*, *Listeria monocytogenes*, and *Bacillus cereus*). The following three strains that showed antimicrobial activity against all indicators were identified: *Enterococcus faecalis* DH9003 and DH9012, and *Lactococcus lactis* DH9011. Their antimicrobial products exhibited typical bacteriocin characteristics, such as heat stability and proteinase nature. The bacteriostatic activity of concentrated bacteriocins produced by these LAB was observed at low concentrations (half-minimum inhibitory concentration [MIC50] and 4MIC50), whereas complete inhibition activity against *Listeria monocytogenes* was detected at high concentrations (16MIC50) of the two *Enterococcus faecalis* strains (DH9003 and DH9012). Furthermore, the probiotic potentials of the three strains were investigated and described. The results revealed that none of the strains had hemolytic activity, whereas all: were sensitive to ampicillin (50 mg/mL) and streptomycin sulfate (100 mg/mL); were resistant to bile, artificial simulated intestinal tract, and gastric juice at different pH levels (2.5, 3.0, 3.5); and had β-galactosidase activity. Furthermore, all strains exhibited an auto-aggregating phenotype, with self-aggregation ranging from 30% to 55%. DH9003 and DH9012 co-aggregated well with *Listeria monocytogenes* and *Escherichia coli* (52.6% and 63.2%, 68.5% and 57.6%, respectively), whereas DH9011 co-aggregated poorly with *Listeria monocytogenes* (15.6%) and did not co-aggregate with Escherichia coli. Furthermore, our results showed that all three isolates exhibited strong antibacterial activity, tolerance to bile and simulated gastrointestinal environments, adhesion capability, and safety. Finally, DH9003 was selected and used for gavage in rats. By observing the pathological characteristics of rat intestinal and liver tissue sections, DH9003 showed no harmful effects on the intestine and liver of rats, but rather resulted in a denser and longer intestinal mucosa, as well as improving the intestinal mucosa of rats. Considering their substantial prospective applications, we concluded that these three isolates are potential probiotic candidates.

## 1. Introduction

Probiotics are a class of living microorganisms that improve host metabolism, enhance immunity, regulate intestinal flora, and exert beneficial effects on the host after ingestion [1]. Numerous studies have shown that probiotics have beneficial effects on humans and animal bodies, such as: preventing or reducing the symptoms of inflammatory bowel disease, irritable bowel syndrome, constipation, acute diarrhea associated with antibiotics, hypertension and diabetes; restoring and maintaining the homeostatic balance of intestinal flora in any animal as well as improving the body’s defensive functions [2]; elevating the levels of dietary folate, niacin and riboflavin; reducing lactose intolerance; enhancing systemic immune function and improving allergic reactions. They also have a preventive and therapeutic function against cancer [3], control blood cholesterol levels, and increase the excretion of de-conjugated bile salts from cholesterol [4]. When animals or humans consume probiotics, they can have a probiotic effect on the host if a large number of live bacteria arrive, adhere, and colonize the host’s intestinal tract. As a result, probiotics must be safe (do not harm the host), functional (colonize the intestinal tract, produce antimicrobial substances, and so on), and feasible (improve the immune system of the host) for use.

Lactic acid bacteria (LAB) are generally recognized as safe microorganisms. They are able to survive under acidic conditions, have high adhesion capacity, and produce antibacterial substances. Therefore, most LAB are regarded as probiotics [5,6]. Relevant studies have screened probiotics from dairy and dairy products of animal origin, among which *Streptococcus thermophilus*, *Lactococcus lactis*, *Enterococcus* spp., and *Lactobacillus* spp. are the most widely studied and commercially concentrated strains of current probiotics [7,8,9,10].

LAB have the ability to ribosomally synthesize antimicrobial peptides known as bacteriocins [11]. Bacteriocins from LAB are normally effective at inhibiting bacteria that are closely related to them. Unlike other antibacterial substances, bacteriocins specifically inhibit and efficiently kill their target bacteria. Some bacteriocins from LAB inhibit a broad spectrum of bacteria, and can prevent food spoilage and suppress the effects of pathogenic bacteria [12]. In addition, bacteriocins from LAB are heat- and acid-base tolerant, easily degradable in vivo, safe, and residue-free [13]. Therefore, screening of bacteriocin-producing LAB and analysis of their potential probiotic properties would be of great importance for the prevention and treatment of intestinal diseases caused by specific pathogenic bacteria.

Goat milk has distinct physicochemical, nutritional, and other health benefits. As a result, goat milk has been identified as the world’s most similar dairy product to human milk. Goat milk has been considered a common resource for bacteriocin screening as a probiotic carrier [14,15]. According to some studies, dairy-derived LAB possesses bacteriocin-like properties as well as medical and probiotic patterns [16,17]. We recently performed a systematic screening of LAB from goat milk. In the current study, three common foodborne pathogenic bacteria (*Listeria monocytogenes*, *Staphylococcus aureus*, and *Bacillus cereus*) were used as indicators of bacteriocin-producing LAB screening strains. LAB with bacteriocin-like properties that can inhibit these foodborne pathogens were characterized molecularly and biochemically. Simultaneously, the potential probiotic properties of LAB screened from goat milk sources were investigated by conducting tolerance and aggregation tests simulating gastrointestinal fluids. The aim of the study was to evaluate the probiotic properties of bacteriocin-producing LAB and provide strain resources for probiotic resource development, with the end goal of providing a basis for their micro-ecological preparation and future industrialization.

## 2. Materials and Methods

### 2.1. Bacterial Strains

*Listeria monocytogenes* LFM2813, *Staphylococcus aureus* LFM3263, *Bacillus cereus* LFM2805, *Lactococcus lactis* LFM2122, and *Escherichia coli* LFM3704 were used for the analyses and preserved at the East China Sea Fisheries Research Institute, Chinese Academy of Fisheries Science.

### 2.2. LAB Screening

Fresh goat milk was obtained from Shaanxi Weinan, China. Approximately 25 mL fresh goat milk was dissolved in 225 mL saline and kept at room temperature for 30 min before gradient dilution under aseptic conditions. Approximately 100 µL of sample solution was coated on de Man, Rogosa, and Sharpe (MRS) (Oxoid, London, UK) and LM17 (0.5% lactose added to M17, Oxoid, UK), mixed with polymyxin B (0.5 g/mL, Sigma-Aldrich, St. Louis, MO, USA), and incubated for 48 h at 30 °C under anaerobic conditions. Single colonies were regrown before being inoculated into a LM17 liquid medium and incubated overnight at 30 °C under anaerobic conditions. Gram staining, peroxidase reaction, and pH measurement were used to screen the LAB, and all strains obtained in the study were stored in 13% (*v*/*v*) glycerol (Apexbio, Shanghai, China) at −80 °C until further analysis.

### 2.3. Determination of Antimicrobial Activity

Three foodborne pathogenic bacteria were used as the indicator strains, namely *Listeria monocytogenes* LFM2813, *Staphylococcus aureus* LFM3263, and *Bacillus cereus* LFM2805, and all isolated LAB were used for screening. The agar diffusion method was used for antimicrobial activity screening as described [18].

### 2.4. Determination of Bacteriocin-like Substance Stability

The isolated LAB with inhibitory activity were incubated overnight at 30 °C and then centrifuged for 30 min at 10,000 rpm. The supernatant was collected, and the residual bacteria were removed using a 0.22-µm membrane. The supernatant was placed in a water bath at 100 °C for 10 min, and inhibition of the indicator bacteria was determined by the microdilution method using a 96-well microtiter plate with reference to the method as described [18].

All the LAB isolates were incubated overnight at 30 °C, 3 μL of the LAB isolates were added to the medium with *Listeria monocytogenes* LFM2813 according to the agar diffusion method, and 1 μL of different enzyme reagents (20 μg/mL proteinase K and 1 mg/mL trypsin) was added dropwise next to the LAB isolates as described [18]. The ability of the enzymes to decompose the inhibitory substances was determined by adding of 3 μL of the LAB isolates without the above treatment as a blank control.

### 2.5. Genomic DNA Extraction, Polymerase Chain Reaction (PCR) Amplification, and Strain Identification

Genomic DNA was extracted using a GenElute Bacterial Genomic DNA Kit (Sigma-Aldrich, St. Louis, MO, USA) according to the manufacturer’s instructions. PCR amplification was performed using the genomic DNA of the LAB isolates with bacterial inhibitory activity as a template, using Taqase with primers 12F: 5′-AGGGTTGCGCTCGTTG-3′ and 15R: 5′-TACGGGAGGCAGCCAG-3′ to establish a 50-µL amplification system for strain identification.

Repetitive-sequence-based PCR (rep-PCR) was performed as described [19]. The genomic DNA of all the LAB isolates with inhibitory activity was used as a template for amplification by establishing a 20-μL amplification system using BOX:5′-CTACGGCAAGGCGACGCTGACG-3′ as the primer to differentiate genetic profiles of the LAB isolates. PCR amplification products were electrophoresed on 1.5% agarose gels and observed under an ultraviolet lamp.

### 2.6. Identification of Antibacterial Properties

Crude extracts of bacteriocins from the LAB isolates with inhibitory activity were inoculated in 50 mL of BHI broth. The overnight culture was centrifuged at 9000 rpm for at least 30 min. The supernatant of the overnight culture was precipitated using 70% saturated ammonium sulfate and kept at 4 °C for at least 4 h. The mixture was then centrifuged at 10,000 rpm for 30 min. The precipitates, after gentrification, were resuspended with H_2_O containing 0.1% TFA and stored at −18 °C for further use.

The bacteriostatic properties of crude bacteriocin extracts against *Listeria monocytogenes* LFM2813 were determined by the microdilution method, using 96-well microtiter plates according to the methods as described [18], and the growth curves of OD_600nm_-t were plotted.

### 2.7. Probiotics Properties Analysis

#### 2.7.1. Hemolytic Test

The LAB isolates with inhibitory activity were analyzed for hemolytic activity as described [20]. Briefly, all the LAB isolates were grown on rabbit serum plates and incubated with *Bacillus cereus* LFM2805 as a control for 24 h at 30 °C, after which they were observed.

#### 2.7.2. Determination of Antibiotic Tolerance

Antibiotic resistance of all the LAB isolates with inhibitory activity was determined using the agar diffusion method as described [21].

#### 2.7.3. Determination of Sugar Metabolism Capacity

The sugar metabolism capacity of all the LAB isolates with bacterial inhibitory activity was determined using an API 50CH (Biomerieux, Marcy I’Etoile, France) bacterial sugar metabolism kit. Using the kit, samples were incubated at 30 °C for 24 h, and color changes were then recorded, compared, and determined after 48 h.

#### 2.7.4. Determination of Tolerance to Bile Salts

Bile salts of different mass fractions (0, 0.2%, 0.4%, 0.6%, 0.8%, and 1%) were added to the Brain heart infusion (BHI, Oxoid, Hampshire, UK) liquid medium and sterilized. All the LAB isolates with bacterial inhibitory activity were inoculated in the BHI liquid medium at 4% of the inoculum, and their optical density (OD) at 600 nm was determined after overnight incubation at 30 °C under aerobic conditions. The bile salt tolerance of the indicator bacteria with bacterial inhibitory activity was determined using the method described [21].

#### 2.7.5. Determination of Tolerance to Simulate Gastrointestinal Fluids

The tolerance of the screening LAB isolates to simulate gastrointestinal fluids was determined as described [17]. The LAB isolates were inoculated overnight at 30 °C in BHI liquid medium and then centrifuged at 3500 rpm for 15 min, after which the precipitated cells were collected, washed twice with sterile saline, and stored after thorough removal of the medium. About 50 mL of sterile saline (containing 1% pepsin: pH 2.5, 3.0, or 3.5) was used to simulate gastric juice. About 100 mL of 0.68% sterile KH_2_PO_4_ solution (containing 1% trypsin) was used to simulate intestinal fluid. The prepared cells were resuspended in the artificial gastrointestinal fluid and incubated at 37 °C, and the OD_600nm_ values of the culture solution were measured every 1 h.

#### 2.7.6. Analysis of Bacterial Self-Aggregation and Co-Aggregation

All the LAB isolates with inhibitory activity, namely *Listeria monocytogenes* LFM2813 and *Escherichia coli* LFM3704, were collected according to the method as described [17] and resuspended in phosphate-buffered saline (PBS, pH 6.8) to a bacterial concentration of 10^8^ CFU/mL. This method of analysis of the self-aggregation and co-aggregation of the LAB isolates with bacterial inhibitory activity was carried out as described [17].

#### 2.7.7. Determination of β-Galactosidase Activity

The β-galactosidase activity of all the LAB isolates with bacterial inhibitory activity was determined using a β-galactosidase activity assay kit and expressed as U/10^4^ cells.

### 2.8. Animal Experiment

A total of 16 male C57 rats (four weeks old, average weight: 15–17 g) were obtained from Fudan University (Shanghai, China). All rats were kept separately at 25 °C with 50–60% relative humidity. All of the rats were starved for 12 h prior to the experiment, and they were divided into two groups (*n* = 8 per group). As the test group, rats were fed a LAB suspension at a concentration of 0.2 mL per 10g-rat weight, while the control group was fed sanitary saline. After four weeks of treatments, all of the rats were killed in accordance with ethical protocols and procedures [22]. The Animal Care and Use Committee at Fudan University approved the animal experiments.

### 2.9. Tissue Section Analysis

Rats were executed by cervical dislocation at the end of the feeding experiment (all rats were starved for 12 h the night before execution). After dissection, liver and small intestine tissue of appropriate size was cut; washed with sterile saline and fixed in formaldehyde fixative (formaldehyde:PBS = 1:9); trimmed; dehydrated; embedded; and sectioned as described [23]; stained with hematoxylin-eosin staining; and the liver and intestinal tissue structures were observed under a light microscope after blocking the sections.

### 2.10. Data Analysis

The experimental data were analyzed using Microsoft software and plotted using Origin 2018. The sequences obtained after the identification of PCR amplification products were determined by comparison with the BLAST search system database in NCBI.

## 3. Results and Discussion

### 3.1. Screening and Identification of LAB with Antimicrobial Activity

The microbial quality of goat milk was analyzed using Gram staining (positive), peroxidase reaction (negative), and pH of LAB broth (4.0). In total, 46 LAB were isolated in this study. Among all isolates, three candidates exhibited antimicrobial activity against a panel of three different pathogenic strains that are normally found in milk, food, and the environment. Subsequently, the three isolates were genotyped by 16S rDNA sequencing. According to the sequencing results, the isolates were *Enterococcus faecalis* DH9011 and DH9012 and *Lactococcus lactis* DH9003, with 99% sequence identity.

Dairy products are the common resources for LAB where lactoccoci, lactobacilli, enteroocii, and other strains are prevalent as active producers. Screening for bacteriocins from dairy-derived LAB is the common strategy, and many bacteriocins have been reported using the strategy [24,25,26,27]. Like most bacteriocins, lactococci bacteriocins (except nisin, garvicin KS, and garvicin ML) have relatively narrow inhibition spectra. In our study, three LAB strains showed antimicrobial activity against three different genera of indicator strains, indicating that the three LAB strains may have broad inhibition spectra.

Bacteriocins derived from enterococci and lactococci strains have been extensively researched [28,29]. Lactoccocins, such as lactococcin A, lactococcin G, and others, typically have relatively narrow inhibition spectra, primarily against close species [30,31]. Enterocins inhibited a wide range of spoilage bacteria and foodborne pathogens, including *Listeria monocytogenes*, *Bacillus cereus*, *Staphylococcus aureus*, and *Clostridium* sp. [32,33]. There have also been some studies on the antagonistic activities of enterocins against Gram-negative bacteria, fungi, yeasts, and viruses [34,35]. Several bacteriocins from *Enterococcus faecalis* have been well studied genetically and biochemically [36,37]. To avoid finding identical and well-known *Enterococcus faecalis* bacteriocins, *Enterococcus faecalis* producing strains of enterocin A, enteriocin P, and enterolysin A were used against DH9003 and DH9012 due to the self-immunity defense of bacteriocins. The antagonistic activity of DH9003 and DH9012 against the three enterocin-producing strains was observed, indicating that DH9003 and DH9012 most likely produce different bacteriocins. Since many enterocins have been discovered and studied, more research on production extraction and analysis from *Enterococcus faecalis* DH9003 and DH9012 is required.

### 3.2. Determination of Antimicrobial Stability of Bacteriocin-Producing LAB Stains

LAB can produce bacteriocins, organic acids, and other bacteriostatic compounds [38]. To reduce the influence of other substances, the stability of bacteriostatic substances produced by the indicator bacteria under the influence of heat and proteases was investigated because bacteriocins have high thermal stability and are easily decomposed by proteases. The stability of bacteriocins produced by the three LAB strains after heat and enzyme treatments is shown in Table 1. The results showed that the bacteriocins of the three LAB strains retained inhibitory activity after heat treatment but were all inactivated by protease treatment. This result confirmed that the inhibitory substances produced by the three LAB strains had the basic properties of bacteriocins.

### 3.3. Rep-PCR

Bacterial genomic rep-PCR fingerprinting is a highly stable technique that allows the comparison of bacterial strain affinities, diversity, and differences using PCR amplification products [39]. To determine whether the bacterial genotypes of the same genus were consistent, rep-PCR analysis was performed by PCR amplification of genomic repeat sequences of DNA from the screened LAB, using BOX primers to determine their corresponding genetic profiles for further differentiation. The electrophoretic profiles of the rep-PCR amplification products of the three strains of bacteriocin-producing LAB are shown in Appendix A. The experimental results showed that the rep-PCR amplification products of the three strains of bacteriocin-producing LAB differed significantly, indicating genetic homology for the three strains.

### 3.4. Fermentation Profiles of the Three Bacteriocin-Producing Strains

Table 2 shows the fermentation capacity of the three bacteriocin-producing LAB strains. The ability of the three LAB strains to metabolize carbohydrates such as glucose was comparable to that of the other LAB strains. However, DH9011 lacked the ability to metabolize L-arabinose, sorbitol, α-methyl-d-mannoside, and melibiose; DH9003 lacked the ability to metabolize D-xylose, sorbitol, and amygdalin. DH9012 lacked the ability to metabolize D-xylose, α-methyl-d-mannoside, melibiose, and D-trehalose anhydrous. The fermentation abilities of the three bacteriocin-producing LAB strains differed, as shown by the results of sugar metabolism tests. Consequently, the homology similarity of these strains was determined to be low. Overall, all three strains showed different genotype and phenotype, indicating that all three strains may produce different bacteriocins. Therefore, all three strains were used in subsequent analyses.

### 3.5. Bacteriostatic Properties of the Three Bacteriocin-Producing Strains

Figure 1 shows the inhibition curves of the three bacteriocin producers against *Listeria monocytogenes*, illustrating how these three bacteriocin-producing strains killed the pathogens. The results showed that all three strains of bacteriocin producers inhibited *Listeria monocytogenes* to some extent when added at the MIC50 and 4MIC50; however, as time progressed, *Listeria monocytogenes* re-grew. This suggested that bacteriocins at low concentrations (MIC50 and 4MIC50) could inhibit the growth of *Listeria monocytogenes*. When the bacteriocin concentrations of *Enterococcus feacalis* DH9003 and DH9012 reached 16MIC50, *Listeria monocytogenes* did not grow, indicating that the bacteriocins produced by DH9003 and DH9012 at high concentrations could completely inhibit *Listeria monocytogenes*’ growth.

Bacteriocins from LAB generally kill target cells by forming pores in their membranes [40]. The formation of pores in the membrane allows low-weight molecules to leak through, eventually resulting in dissipation of the proton motive force. According to recent studies, bacteriocins are only effective against protein-containing cellular membranes [41]. In the previous findings, bacteriocins may use specific membrane proteins to confer sensitivity [41]. In the current study, a portion of the cells was killed by low concentrations of bacteriocins, although DH9003, DH9011, and DH9012 began to grow again after 6 h, 6 h, and 10 h, respectively. Similar results were obtained for nisin, pediocin, and other well-known bacteriocins. However, relatively high concentrations of bacteriocins are required to eradicate *Listeria monocytogenes*. Our findings indicated that high amounts of bacteriocin-producing strains were capable of fighting pathogens.

### 3.6. Characterization of the Probiotic Effects of the Three Bacteriocin-Producing Strains

Probiotic bacteria in the intestinal tract play important roles, and are able to withstand the destructive ability of small intestinal fluid to survive and proliferate in the stomach and intestinal tract. At the same time, bile salts in the duodenum have an inhibitory effect on exogenous bacteria, and the probiotic bacteria must also have a high tolerance to bile salts to maintain a sufficient number of viable bacteria and exert their probiotic effects. Table 3 shows the probiotic properties of the three LAB. The results showed that all three LAB were able to maintain a high number of viable bacteria in the bile salt environment; therefore, we concluded that all three were able to adapt to the intestinal bile salt environment. In addition, another bottleneck for the survival and proliferation of LAB in the gastrointestinal tract of animals is that they must be able to withstand the destructive effects of the low acidity of gastric juice, containing pepsin, and intestinal juice, containing trypsin. The results of our experiment showed that all three LAB remained viable in these environments, suggesting that they could tolerate the gastrointestinal and gastric fluid environments under different pH conditions (2.5, 3.0, and 3.5).

Self-aggregation and co-aggregation are important properties of probiotics, corresponding to their ability to form biofilms on intestinal epithelial cells, and the combination of probiotics and intestinal pathogens contributes to the killing of pathogenic bacteria by antimicrobial substances [42]. The experimental results showed that the three LAB strains had good self-aggregation, ranging from 30% to 55%, and the co-aggregation tests with *Listeria monocytogenes* and *Escherichia coli* were relatively high for *Enterococcus faecalis* DH9003 and DH9012 (52.6% and 63.2%, 68.5% and 57.6%, respectively), whereas *Lactococcus lactis* DH9011 had a 15.6% co-aggregation rate with *Listeria monocytogenes* and did not aggregate with Escherichia coli, suggesting that *Enterococcus faecalis* DH9003 and DH9012 may have better adhesion potential.

Lactose intolerance is caused by the lack of β-galactosidase in the duodenal lining, and all three strains of bacteriocin-producing LAB were found to have β-galactosidase activity, which contributes to lactose intolerance. In addition, all three LAB strains are non-hemolytic (γ-hemolysis) and safe for humans. Finally, in our analysis of antibiotic tolerance, all three LAB strains were found to be sensitive to ampicillin (50 mg/mL) and streptomycin sulfate (100 mg/mL) and insensitive to chloramphenicol (2.5 mg/mL), nalidixic acid (10 mg/100 mL), and furantoin (10 mg/100 mL).

The digestive tract is the primary site of interaction between microorganisms and the host’s immune system, allowing the survival of intrinsic microorganisms while defending against pathogens. Under normal conditions, probiotics and pathogenic bacteria in the intestinal tract rely on each other to achieve a state of balance, forming a flora barrier of the intestinal mucosa.

The FAO/WHO criteria identify three basic characteristics that must be met by probiotics: tolerance to the selective environment of the gastrointestinal mucosa, adhesion to the host’s intestinal wall cells, and secretion or breakdown products as antimicrobial substances [43]. The three LAB strains isolated from goat milk in this experiment were shown to produce bacteriocins that can inhibit common food-borne pathogenic bacteria, are able to tolerate the environment of the gastrointestinal tract, and have good adhesion ability, among other characteristics. At the same time, these three strains are safe (non-hemolytic), controllable (nonresistant), and have potential probiotic qualities.

### 3.7. Observation of Rat Intestine Section

The liver is the foundation for the health of the remainder of the organ system and is crucial in regulating a stable internal environment and breaking down possibly harmful substances [44]. Therefore, it is feasible to accurately identify whether *Enterococcus faecalis* DH9003 has a negative impact on the organism by evaluating the histopathological changes in rat liver. Figure 2a,b show the microscopic observation of liver sections of rats in the blank control and treated groups, and it was found that the liver tissues of rats in both the blank control and *Enterococcus faecalis*-treated groups were uniform in structure and did not produce lesions, indicating that *Enterococcus faecalis* DH9003 did not have harmful effects on the liver of rats.

The primary organ for nutrition absorption and tissue blockage is the small intestine. Approximately 70% of immune cells and immunoglobulins are found there, making it the body’s biggest immune organ [45]. As a result, one of the factors used to determine the health of the organism is the intestine’s normalcy. We gave the rats *Enterococcus faecalis* DH9003 to ingest in order to examine the capacity of prospective probiotic candidates to affect the intestinal tract of the rats. We then measured the effects using tissue sectioning. Figure 2c,d show the microscopic observation of the intestinal sections of rats in the blank control group and the *Enterococcus faecalis*-treated group. The intestinal epithelial cells in both groups were closely arranged and relatively intact in structure, the intestinal glandular epithelial cells were tightly bound to the basement membrane, and the intestinal villi were intact, with no obvious abnormalities. It is noteworthy that the intestinal villi of rats in the *Enterococcus faecalis* DH9003-treated group were more densely arranged, and the length of intestinal villi increased compared with that of the blank control group, which indicates that *Enterococcus faecalis* DH9003 did not have a negative effect on the intestinal tract of rats, and also had a certain improving effect on the intestinal mucosal barrier of rats.

## 4. Conclusions

In this study, 46 LAB strains were isolated from fresh goat milk and confirmed by Gram staining (positive), peroxidase reaction (negative), LAB broth pH (≤4.0), and ability to metabolize large amounts of carbohydrates. Three out of the 46 LAB strains inhibited *Listeria monocytogenes*, *Staphylococcus aureus*, and *Bacillus cereus*. After 16S rDNA molecular identification, these three LAB strains were identified as two strains of *Enterococcus faecalis* and one strain of *Lactococcus lactis*. The bacteriocins produced by these three strains prevented the growth of *Listeria monocytogenes* at low concentrations (MIC50, 4MIC50), and only the bacteriocins produced by DH9003 and DH9012 completely inhibited *Listeria monocytogenes* at high concentrations (16MIC50). In addition, the three strains of bacteriocin-producing LAB were safe (non-hemolytic) and controllable (sensitive to ampicillin [50 mg/mL] and streptomycin sulfate [100 mg/mL]). Additionally, these strains were able to adapt to environments with bile salts, as well as simulated intestinal and different gastric fluid conditions (pH 2.5, 3.0, and 3.5, respectively), with good self-aggregation (30–55%). DH9003 and DH9012 showed good co-aggregation with *Listeria monocytogenes* and *Escherichia coli*, with a concentration range of 50–70%, whereas DH9011 showed good co-aggregation with DH9011, poor co-aggregation with *Listeria monocytogenes* (15.6%), and no aggregation with *Escherichia coli*. Finally, the three strains also showed high β-galactosidase activity. Furthermore, the experimental results revealed that the three LAB strains had strong antibacterial ability and a good ability to mimic gastrointestinal fluid tolerance and adhere to the intestine, consistent with the potential characteristics of probiotics. In addition, the results of intestinal and liver tissue sections showed that the intake of *Enterococcus faecalis* DH9003 resulted in a denser and longer intestinal mucosa in rats, which had a certain improvement effect on the intestinal mucosa, and did not cause harmful effects or damage to the liver. Overall, our findings provided a theoretical basis for further development of probiotic resources and subsequent in vivo experiments with probiotics.

## Figures and Tables

**Figure 1 microorganisms-11-00849-f001:**
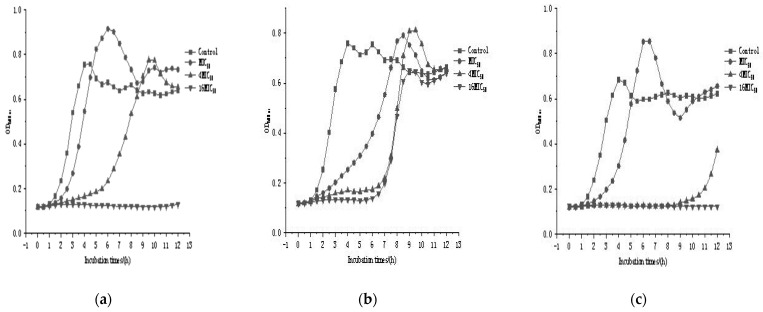
Growth curve of three bacteriocin producers from lactic acid bacteria against Listeria monocytogens at different half-minimum inhibitory concentrations. (**a**) DH9003; (**b**) DH9011; (**c**) DH9012.

**Figure 2 microorganisms-11-00849-f002:**
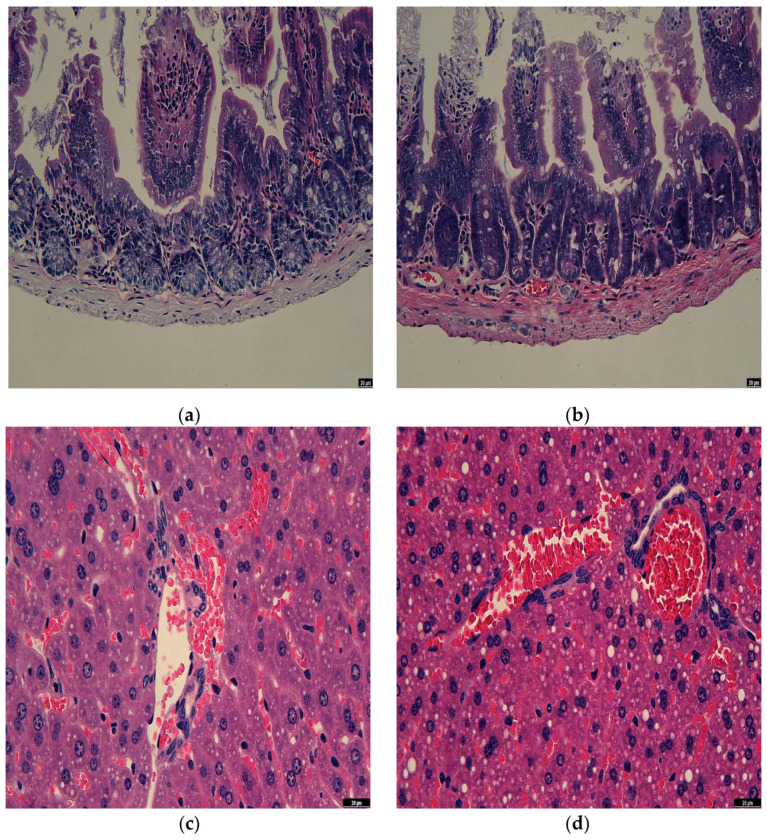
Results of rat tissue sections. (**a**) small intestine without *Enterococcus faecalis* DH9003 treatment; (**b**) small intestine with *Enterococcus faecalis* DH9003 treatment; (**c**) liver without *Enterococcus faecalis* DH9003; (**d**) liver with *Enterococcus faecalis* DH9003 treatment.

**Table 1 microorganisms-11-00849-t001:** Antimicrobial stability of bacteriocins produced by LAB under different treatments.

Samples	Heat Treatment	Enzyme Treatments	
	100 °C, 10 min	Trypsin	Proteinase K
DH9003	+	-	-
DH9011	+	-	-
DH9012	+	-	-

Note: + indicates activity, - indicates inactivity.

**Table 2 microorganisms-11-00849-t002:** Comparisons on fermentation profiles of three bacteriocins producers from lactic acid bacteria.

Sugar Components	DH9003	DH9011	DH9012
L-Arabinose	+	-	+
D-xylose	-	+	-
Sorbitol	-	-	+
α-methyl-d-mannoside	+	-	-
Amygdalin	-	±	-
Melibiose	+	-	-
D-Trehalose anhydrous	+	+	-

Note: + indicates metabolizable, - indicates not metabolizable, ± indicates weak metabolizability. Other components not marked in the table indicate the same metabolic capacity.

**Table 3 microorganisms-11-00849-t003:** Probiotic properties of three bacteriocins producers from lactic acid bacteria.

Properties	DH9003	DH9011	DH9012
Bile salt tolerance	+	+	+
Artificial gastric juice (pH 2.5)	+	+	+
Artificial gastric juice (pH 3.0)	+	+	+
Artificial gastric juice (pH 3.5)	+	+	+
Artificial intestinal fluids	+	+	+
Self-aggregation (%)	52.3	46.9	30.5
Co-aggregation with *Listeria monocytogenes* (%)	52.6	15.6	68.5
Co-aggregation with *Escherichia coli* (%)	63.2	N	57.6
Hemolysis	γ-hemolysis	γ-hemolysis	γ-hemolysis
β-Galactosidase activity (U/10^4^ cell)	0.678	0.611	0.596
Ampicillin (50 mg/mL)	S	S	S
Streptomycin sulfate (100 mg/mL)	S	S	S
Chloramphenicol (2.5 mg/mL)	R	R	R
Nalidixic acid (10 mg/100 mL)	R	R	R
Furantoin (10 mg/100 mL)	R	R	R

Note: + indicates tolerance, N indicates non-aggregation, S indicates sensitivity, and R indicates resistance.

## Data Availability

Not applied.

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
