# Peer review of "Screening and Identification of Goat-Milk-Derived Lactic Acid Bacteria with Bacteriocin-like Activity and Probiotic Potentials"

_microorganisms, 2023, doi:10.3390/microorganisms11040849_

Round 1

Reviewer 1 Report

The manuscript presents a study on the identification of three lactic acid bacteria from goat milk with bacteriocin-like activity. The identified three LAB have the probiotic potentials. Overall the manuscript is well structured. However, there are some issues that need to be addressed.

The abstract should be a total of about 200 words maximum. Please revise the abstract and should follow the structure of background, methods, results and conclusion.

The significance of the study should be further discussed. The application of these identified three LAB and the advantages of using these strains as probiotics compared with other current strains should also be discussed.

For the Section 2 materials and methods, suggest re-organizing the subtitles based on the content in Section 3. Can create sub-titles such as 2.2.1 under 2.2. Put all the related methods under a major analysis.

Author Response

Dear editor,

I would like to express our gratitude to the reviewers for their comments. Your comments help the manuscript much more clearer. I made the modified verision according to your comments.

1.The abstract should be a total of about 200 words maximum. Please revise the abstract and should follow the structure of background, methods, results and conclusion.

A: Thanks for your comments. the abstract has been changed, but not too much. first we would like to address the results as much as we have done, second we saw several manuscripts had long abstracts,

2. The significance of the study should be further discussed. The application of these identified three LAB and the advantages of using these strains as probiotics compared with other current strains should also be discussed.

A: Thanks for your comments. We have added some information in the manuscript, such as L54-L59, and some discussion at L225-L240.

3. For the Section 2 materials and methods, suggest re-organizing the subtitles based on the content in Section 3. Can create sub-titles such as 2.2.1 under 2.2. Put all the related methods under a major analysis.

A:  We have re-organized the Section 2 part accoring to your comments.

Reviewer 2 Report

This manuscript screen and identify goat milk-derived lactic acid bacteria. Three strains are found to have the bacteriocin-like activity and probiotic potentials. The study is interesting and has substantial prospective application. I would like to recommend this manuscript for publishing after minor revision. Some reasons were presented as follows.

1.      The research background should be supplemented in the abstract.

2.      The main research methods should be mentioned in the abstract.

3.      Please add the age of the C57 rats.

4.      Line 158, 168: “OD600nm” need to be modified to “OD600nm

5.      In 3.1 part, the authors conducted an experiment on antibacterial activity, please supplement the corresponding result chart, such as the result of the agar diffusion method.

Author Response

Dear editor,

I would like to express our gratitude to the reviewers for their comments. Your comments help the manuscript much more clearer. I made the modified verision according to your comments.

1 and 2 about the abstract.

A: The absctract of the manuscript has been changed. 

3. Please add the age of the C57 rats.

A: The age of the C57 rats has been added in L187.

4. Line 158, 168: “OD600nm” need to be modified to “OD600nm

A: Subscript at 600nm has been applied.

5.In 3.1 part, the authors conducted an experiment on antibacterial activity, please supplement the corresponding result chart, such as the result of the agar diffusion method.

A: The following results at section 3.5 has showed the antimicrobial activity of three strains against Listeria monocytogenes, indicating that all three strains had bacteriocin properties. therefore, we did not apply the agar diffusion results in the manuscript.

Reviewer 3 Report

This is an interesting study and good work

I have some minor comments to be addressed

Page 3 the use of indicator bacteria and indicator strains for LAB and test organisms is misleading to the respective reader. Please use other terms

Section 2.5 indicate the used primers for 16SrRNA sequence and indicate the genes for the primers 12F 15R ? Clarify and also clarify the data for BOX primers

Section 2.8 API is used for other sugars not only glucose metabolism capacity   please clarify and correct

L150 correct to stimulate

L160, it is not clear the author used extract or bacterial culture??Clarify this section 2.11

Section 2.12 is not clear. Please rephrase correctly and describe in detail to be understandable

L214 correct genus to genera

L252 is the genetic homology for the three strains OK?

Correct d and L formatting for sugars at L 258-261

Correct the formatting of microorganisms names at L422/424/426/429/432/435/438457/465/470/473/478/485/495/500/503/515/523……etc

Author Response

Dear editor,

First of all, I would like to express our gratitude to the reviewers for their comments. Your comments help the manuscript much more clearer. I made the modified verision according to your comments.

1.Page 3 the use of indicator bacteria and indicator strains for LAB and test organisms is misleading to the respective reader. Please use other terms.

A: all the terms have been changed.

2. Section 2.5 indicate the used primers for 16SrRNA sequence and indicate the genes for the primers 12F 15R ? Clarify and also clarify the data for BOX primers

A: I have clarified the data using different primers.

3.Section 2.8 API is used for other sugars not only glucose metabolism capacity   please clarify and correct

A: I have changed the glucose into sugar in Section 2.8

4. L150 correct to stimulate

A: The word "stimulated" has changed into "stimulate"

5.L160, it is not clear the author used extract or bacterial culture??Clarify this section 2.11

A: section 2.11 has been changed.

6.Section 2.12 is not clear. Please rephrase correctly and describe in detail to be understandable

A: section 2.12  has been changed.

7.L214 correct genus to genera

A: I have changed the "genus" into "genera" in L214.

8.L252 is the genetic homology for the three strains OK?

A: I have added the "for the three strains" in new version L268.

9.Correct d and L formatting for sugars at L 258-261

A: I have changed the d and l formatting for the sugars in the correct way.

10. Correct the formatting of microorganisms names at L422/424/426/429/432/435/438457/465/470/473/478/485/495/500/503/515/523……etc

A: All the formatting of microorganisms names have been changed